# Activation of Peroxymonosulfate by Co-Ni-Mo Sulfides/CNT for Organic Pollutant Degradation

**DOI:** 10.3390/molecules29153633

**Published:** 2024-07-31

**Authors:** Shihao You, Jing Di, Tao Zhang, Yufeng Chen, Ruiqin Yang, Yesong Gao, Yin Li, Xikun Gai

**Affiliations:** 1School of Biological and Chemical Engineering, Zhejiang University of Science and Technology, Hangzhou 310023, China; 212203817017@zust.edu.cn (S.Y.); zhangtaoshd@163.com (T.Z.); 211122050045@zjut.edu.cn (Y.C.); yruiqin@163.com (R.Y.); 2China Construction Eco-Environmental Group Co., Ltd., Beijing 100037, China; gaosong8208@163.com; 3Ecology and Health Institute, Hangzhou Vocational & Technical College, Hangzhou 310018, China; cherryli1986@126.com

**Keywords:** composite materials, advanced oxidation process, heterogeneous catalysts, metal sulfide, radical

## Abstract

To explore advanced oxidation catalysts, peroxymonosulfate (PMS) activation by Co-Ni-Mo/carbon nanotube (CNT) composite catalysts was investigated. A compound of NiCo_2_S_4_, MoS_2_, and CNTs was successfully prepared using a simple one-pot hydrothermal method. The results revealed that the activation of PMS by Co-Ni-Mo/CNT yielded an exceptional Rhodamine B decolorization efficiency of 99% within 20 min for the Rhodamine B solution. The degradation rate of Co-Ni-Mo/CNT was 4.5 times higher than that of Ni-Mo/CNT or Co-Mo/CNT, and 1.9 times as much than that of Co-Ni/CNT. Additionally, radical quenching experiments revealed that the principal active groups were ^1^O_2_, surface-bound SO_4_^•−^, and •OH radicals. Furthermore, the catalyst exhibited low metal ion leaching and favorable stability. Mechanism studies revealed that Mo^4+^ on the surface of MoS_2_ participated in the oxidation of PMS and the transformation of Co^3+^/Co^2+^ and Ni^3+^/Ni^2+^. The synergism between MoS_2_ and NiCo_2_S_4_ reduces the charge transfer resistance between the catalyst and solution interface, thus accelerating the reaction rate. Interconnected structures composed of metal sulfides and CNTs can also enhance the electron transfer process and afford sufficient active reaction sites. Our work provides a further understanding of the design of multi-metal sulfides for wastewater treatment.

## 1. Introduction

With rapid industrial development, environmental problems have become increasingly severe. Many industrial dyes are discharged into water in various forms, causing significant environmental concerns. Capodaglio [1] pointed out that environmental and health-related impacts, as well as the growing trend towards wastewater reuse, require technologies that can remove and even mineralize pollutants. Advanced oxidation processes (AOPs), such as photocatalysis [2], electrochemical oxidation [3], the Fenton process [4], and Fenton-like oxidation [5], can degrade most contaminants. Currently, numerous efforts are focused on the development of excellent catalysts for AOPs. One method is to develop a catalyst for both electrochemical and Fenton oxidation. Wang et al. [6] synthesized spinel-structured CuCo_2_O_4_ on carbon felt as a bifunctional material for both the catalyst and cathode in an electro-Fenton process using a simple hydrothermal method. They found that CuCo_2_O_4_/CF demonstrated a high RhB removal efficiency and broadened the electro-Fenton process to a near-neutral and weakly alkaline environment. Another strategy involves coupling element doping and the construction of heterostructures to design efficient catalysts. For example, Ning et al. [7] designed S-scheme CdS QD/La-Bi_2_WO_6_ (CS/LBWO) photocatalysts with excellent efficiency, stability, and recyclability for the degradation of RhB. Abdelhamid et al. [8] produced a nanocomposite composed of zeolitic imidazolate frameworks (ZIF-67) and hydrogen titanate nanotubes (HTNTs). The material exhibited the combined effects of the porous structure of ZIF-67 and the adsorption capability of HTNTs. This synergistic effect greatly contributes to the efficient removal of dyes and microplastics. ZIF-8-derived ZnO@N-doped C [9] was also synthesized and used as an excellent photocatalyst for methyl blue and fluorescein. However, these technologies suffer from high costs and low efficiencies. Therefore, there is an urgent need to design suitable catalysts and technologies to achieve high removal efficiencies.

Currently, PS-based AOPs are among the most promising oxidation technologies. The activation of persulfates such as peroxymonosulfate (PMS) for the generation of radical and non-radical species remains challenging [10]. PMS can be activated by chemical catalysis [11], UV radiation [12], electrochemistry [13,14,15], and heat [16,17].

Among the various activation strategies, catalysts based on transition metals such as Co, Fe, Ni, and Cu have been widely investigated due to their low energy consumption and high efficiency [18,19,20,21]. Recently, MoS_2_, a transition-metal sulfide with a two-dimensional layered structure, has gained significant attention as a catalyst or co-catalyst for AOPs. MoS_2_ can either activate PMS or PDS directly or accelerate redox cycles of Fe^3+^/Fe^2+^, Co^3+^/Co^2+^, and Cu^2+^/Cu^+^ in a homogeneous persulfate system [22,23,24,25,26,27,28]. The application of MoS_2_ in AOPs was first reported by Xing, who used a system of Fe^3+^/MoS_2_ to accelerate the classic Fenton reaction [27]. However, the catalytic activity of MoS_2_ in persulfate activation remains unsatisfactory. In addition, homogeneous persulfate systems suffer from high consumption and secondary utilization of metal ions. An effective strategy to improve the activity efficiency is to combine MoS_2_ with other metal compounds, including FeOOH@MoS_2_ [29], MoS_2_/Fe_3_O_4_ [30], MoS_2_/Co_0.75_Mo_3_S_3.75_ [31], CuFe_2_O_4_/MoS_2_ [32,33], and ZnFe_2_O_4_/MoS_2_ [34]. These pioneering studies suggest that it is possible to develop new composites with high PMS activation efficiencies, which could offer great opportunities to bolster the practical applications of AOPs.

Inspired by the above discussion, a sulfide composite catalyst was developed to enhance PMS activation efficiency. Although many transition-metal composites have been reported for catalytic degradation, there are only limited reports on the catalytic degradation of organic pollutants in composites prepared by combining MoS_2_, Co_3_S_4_, and Ni_3_S_4_. Using a simple one-pot hydrothermal method, a ternary sulfide composite grown on the surface of carbon nanotubes (CNTs), denoted as Co-Ni-Mo/CNT, was prepared. The CNTs were decorated with metal sulfides for two main reasons: (i) to enhance the electron transfer process during activation and degradation and (ii) to increase the number of active sites of metal sulfides by utilizing the large surface area of the CNTs. The degradation performance was investigated under various conditions: initial solution pH, dye concentration, catalyst dosage, PMS concentration, and existing ions. The catalyst exhibited superior performance for PMS activation and alleviated metal ion leaching. Finally, the possible degradation mechanism was analyzed using quenching experiments, electron paramagnetic resonance analysis (EPR), and ultra-high-performance liquid chromatography–mass spectroscopy (UPLC-MS).

## 2. Results and Discussion

### 2.1. Characterization of Catalyst

The X-ray diffraction (XRD) patterns of the samples are shown in Figure 1a. The strong peak at 26.1° corresponded to the (002) plane of the CNTs. The diffraction peaks at 16.3°, 26.8°, 31.6°, 33.0°, 38.3°, 47.4°, 50.4°, and 55.3° matched the (111), (220), (311), (222), (400), (422), (511), and (440) planes of NiCo_2_S_4_ (JCPDS No. 20-0782), respectively, indicating the formation of NiCo_2_S_4_. However, there were no significant peaks of MoS_2_, suggesting an amorphous nature or highly dispersed status of MoS_2_ in Co-Ni-Mo/CNT [35,36].

To calculate the metal sulfide mass in Co-Ni-Mo/CNT, a thermogravimetric analysis (TGA) of the composites with and without CNTs (Co-Ni-Mo) was performed in air (Figure 1b). Both samples exhibited weight loss over the entire temperature range. The significant weight loss after 400 °C is ascribed to the conversion of metal sulfides to oxides in the air atmosphere [37]. Nevertheless, a larger weight loss of Co-Ni-Mo/CNT relative to Co-Ni-Mo was observed after 550 °C, attributable to both the combustion of CNT and the conversion of sulfides. Thereby, the mass percent of metal sulfides in Co-Ni-Mo/CNT was calculated to be 52.64% according to the weight loss of both samples at 670 °C.

The N_2_ adsorption–desorption isotherms of Co-Ni-Mo/CNT and pure CNTs were identified as type IV isotherms with H3 hysteresis loops (Figure 1c,d), which illustrates that the pores of Co-Ni-Mo/CNT and CNTs were mainly mesopores and macropores with a slit shape [38]. The specific area and pore volume of Co-Ni-Mo/CNT were 54.52 m^2^/g and 0.4566 cm^3^/g, respectively, slightly smaller than the pristine CNT (91.42 m^2^/g and 0.6166 cm^3^/g), which could be attributed to the blockage of CNTs by the sulfides. To further illustrate the structure and morphology of the Co-Ni-Mo/CNT catalyst, scanning electron microscopy (SEM) and transmission electron microscopy (TEM) images were obtained (Figure 2a,b). The SEM images revealed that the catalyst comprised nanoparticles, nanosheets, and CNTs. The high-resolution TEM image (Figure 2c) showed lattice fringe spacings of 0.615, 0.286, and 0.236 nm, corresponding to the (002) plane of MoS_2_ and the (311) and (400) planes of NiCo_2_S_4_, respectively. Elemental mapping analysis using energy-dispersive X-ray spectroscopy (EDX) (Figure 2d–i) confirmed the coexistence of Mo, Co, Ni, S, and C in the composite. S was homogeneously distributed throughout the nanosheets and nanoparticles. More Mo was observed on the surface of nanosheets than on the sites of the nanoparticles, whereas Ni and Co were gathered in the nanoparticle field, and fewer were dispersed on the surface of the CNTs. This distribution suggests that MoS_2_ forms nanosheets, while NiCo_2_S_4_ is mainly present in nanoparticles. Both are entangled in the CNT network, resulting in an interconnected structure. The TEM and XRD results demonstrated that both NiCo_2_S_4_ and MoS_2_ were present in the composite.

To further investigate the surface elemental compositions and complex chemical states, X-ray photoelectron spectroscopy (XPS) analysis was performed on Co-Ni-Mo/CNT, Co/CNT, Ni/CNT, and Mo/CNT. As shown in Figure 3a, the high-resolution Co 2p spectra for both Co/CNT and Co-Ni-Mo/CNT can be deconvoluted into two pairs of spin–orbit doublets with two satellites. The peaks of Co/CNT at 779.0 and 793.9 eV were fitted to 2p2/3 and 2p1/2 of Co^3+^, respectively, whereas those at 782.3 and 798.2 eV corresponded to 2p2/3 and 2p1/2 of Co^2+^ [39], respectively. Similarly, the peaks of Co-Ni-Mo/CNT were divided into Co^3+^ located at 778.8 and 793.5 eV and Co^2+^ located at 781.2 and 796.7 eV, respectively. The proportion of Co^3+^ in (Co^3+^+Co^2+^) for Co/CNT was 45.4%. It decreased to 36.0% for Co-Ni-Mo/CNT. The high-resolution Ni 2p spectrum in Figure 3b illustrates the peak shift between Ni/CNT and Co-Ni-Mo/CNT. The six peaks located at 852.6, 869.7, 860.7, 855.7, 872.9, and 878.5 eV were indexed to Ni 2p3/2 (Ni^2+^), Ni 2p1/2 (Ni^2+^), satellite peaks, Ni 2p3/2 (Ni^3+^), Ni 2p1/2 (Ni^3+^), and satellite peaks, respectively. All the peaks were positively shifted to 853.6, 870.7, 860.9, 856.4, 873.9, and 879.5 eV when Co and Mo were compounded in the catalyst. In addition, the proportion of Ni^3+^ ions in (Ni^3+^+Ni^2+^) decreases from 61.0% to 58.5%. The spectrum of Mo3d in the Mo-CNTs was deconvoluted into two pairs of spin–orbit doublets, as shown in Figure 3c. The peaks at 229.4 and 232.6 eV could be indexed to Mo^4+^ species in MoS_2_, whereas the peaks at 233.2 and 235.8 eV were characteristics of Mo^6+^ species in MoO_3_, owing to the oxidation exposed to air during preparation [40]. Moreover, the peak located at 226.6 eV was assigned to S 2s, suggesting the formation of metal–S bonds in the catalyst. In Co-Ni-Mo/CNT, Mo 3d spectra were deconvoluted into three groups of spin–orbit doublets, namely Mo^4+^ (228.9 and 232.1 eV), Mo^5+^ (230.2 and 233.2 eV), and Mo^6+^ (233.3 and 236.1 eV), and a peak located at 226.5 eV belonged to S 2s. The peaks of Mo^5+^ imply the partial oxidation of Mo^4+^ [41]. Notably, the Mo 3d peaks of Co-Ni-Mo/CNT shifted to lower binding energies. On the basis of the up-shift of Ni 2p peaks and the down-shift of Mo 3d peaks, we conclude that the strong electronic connection between MoS_2_ and NiCo_2_S_4_ forms in the composite [42,43,44]. In addition, the proportion of Mo^4+^ in Co-Ni-Mo/CNT was 50.3%, compared to 79.4% in Mo/CNT, again suggesting electronic migration between Mo^4+^ and NiCo_2_S_4_ [45]. Electronic migration between metal sulfides is beneficial for enhancing the conductivity and catalytic activity [44]. This would affect the binding energies of intermediates and improve the catalytic rates.

### 2.2. Degradation Effect of Co-Ni-Mo/CNT

To assess the synergistic effects between the different metal sulfides, a series of experiments were conducted to evaluate the removal efficiency of Rhodamine B under identical conditions with different catalysts. Prior to degradation, adsorption experiments were conducted to determine the adsorption equilibrium of each catalyst, which was reached within 60 min. Figure 4a shows the Rhodamine B adsorption profiles for the different catalysts. After 60 min of adsorption, the highest removal rate (2.9%) was obtained for Co-Ni-Mo/CNT, indicating that the adsorption capacities of the catalysts were relatively low. To eliminate the effect of adsorption completely, the solutions of the contaminant and catalyst were stirred continuously until adsorption equilibrium before degradation. The degradation performance of the PMS is illustrated in Figure 4b. Compared to the PMS system without a catalyst, all other systems with catalysts exhibited superior degradation performances. Among these systems, Mo/CNT exhibited a relatively poor PMS activation ability, achieving 33.5% degradation within 30 min (63.4% within 60 min), which is comparable to previous reports [26,30]. When binary metal sulfide catalysts such as Ni-Mo/CNT or Co-Mo/CNT were used, 39.9% and 39.4% removal of Rhodamine B was achieved within 30 min, respectively. This suggests that compositing with Ni or Co sulfides could accelerate the catalytic process of MoS_2_ but with limited effects. Interestingly, the ternary metal sulfide catalyst Co-Ni-Mo/CNT exhibited a remarkable synergistic effect with a degradation efficiency of 86.7% within 30 min and 99.3% within 60 min. To illustrate the effect of MoS_2_ on Ni/Co sulfides, the degradation performance of Co-Ni/CNT is also shown. A lower degradation efficiency of 59.6% was achieved within 30 min for Co-Ni/CNT than that for Co-Ni-Mo/CNT, illustrating that the role of MoS_2_ is essential in the activation process. To better reveal the enhancement effect, the degradation curves were analyzed by fitting with pseudo first-order reaction kinetic, and the pseudo first-order degradation rate constants (*k*_obs_) were calculated to be 0.0717 min^−1^, 0.0379 min^−1^, 0.0162 min^−1^, 0.0159 min^−1^, and 0.0155 min^−1^ for Co-Ni-Mo/CNT, Co-Ni/CNT, Co-Mo/CNT, Ni-Mo/CNT, and Mo/CNT, respectively (Appendix A). Therefore, the k_obs_ of the ternary metal sulfide catalyst Co-Ni-Mo/CNT was almost 1.9 times or even higher than those of the binary metal sulfide catalysts Co-Ni/CNT, Ni-Mo/CNT or Co-Mo/CNT and 4.6 times higher than that of Mo/CNT. The degradation results show that the multi-metal combination of Co_3_S_4_, Ni_3_S_2_, and MoS_2_ leads to a unique synergism, effectively accelerating the activation process of PMS.

To further validate the versatility of the Co-Ni-Mo/CNT catalyst, its performance was tested for the degradation of several refractory contaminants with varying structures, as shown in Figure 4c. The results indicate that the catalyst demonstrates high efficiency across different types of contaminants. Anionic azo dyes, such as methyl orange (MO) and Congo red (CR), exhibited notably higher degradation rates and efficiencies compared to cationic dyes like Rhodamine B. Specifically, MO was removed with approximately 99% efficiency within 10 min, while CR was degraded by 95.6% within 40 min. The enhanced performance of anionic azo dyes can be attributed to the favorable adsorption interactions between the contaminants and the catalyst. The initial pH values of the dye and PMS solutions were below 5 due to the hydrolysis of PMS, which makes the Co-Ni-Mo/CNT surface positively charged (pH_pzc_ of 5.63, see Appendix A). This positively charged surface promotes the adsorption of anionic dyes, thereby improving degradation efficiency. Additionally, the catalyst’s performance was tested on other emerging contaminants, including bisphenol A (BPA), phenol, and the antibiotic berberine hydrochloride. The Co-Ni-Mo/CNT catalyst showed high degradation efficiencies for these contaminants as well, with removal rates of 83.2% for BPA, 88.2% for berberine hydrochloride, and 86.7% for phenol within 60 min. These results underscore the practical applicability of Co-Ni-Mo/CNT in various environmental remediation scenarios. Figure 4d presents a schematic of the ion leaching results for the Co-Ni-Mo/CNT, Ni/CNT, Co/CNT, and Mo/CNT catalysts, which provides insights into the stability and potential for metal ion leaching during the catalytic process. The amount of Co dissolution concentration from Co/CNT after the reaction was as high as 11.62 mg/L, significantly surpassing the Co dissolution amount of 1.23 mg/L for Co-Ni-Mo/CNT. Although the Co/CNT catalyst demonstrated a high degradation rate (see Appendix A), it suffers from excessive metal ion leaching, which hampers its practical application. Specifically, the Ni leaching concentration decreased from 4.84 mg/L in Ni/CNT to 2.37 mg/L in Co-Ni-Mo/CNT. The Mo leaching in Co-Ni-Mo/CNT was 1.07 mg/L, compared to 2.32 mg/L for Mo/CNT, indicating a significant reduction in Mo dissolution when combined with other metal sulfides. Thus, the ternary metal sulfide catalyst Co-Ni-Mo/CNT offers substantial improvements in reducing metal leaching while maintaining acceptable degradation performance. This reduction in metal dissolution enhances both environmental sustainability and catalytic stability, making Co-Ni-Mo/CNT a promising candidate for practical wastewater treatment applications.

By comparing different catalysts for the catalytic degradation of Rhodamine B, via the contrast of a variety of aspects, it can be considered that the catalyst Co-Ni-Mo/CNT had a high catalytic effect (Table 1).

### 2.3. Influencing Factors of Rhodamine B Degradation

The effect of the initial pH of the Rhodamine B solution (1.0, 3.0, 6.0, 9.0, and 12.0) on the decolorization efficiency was examined. Figure 5a shows that at pH 1, the degradation efficiency reached 99% within 30 min. As the pH increased, the degradation efficiency gradually decreased until reaching pH 6. However, upon further increasing the pH to 9, a slight enhancement in degradation efficiency was observed. The degradation rate of Rhodamine B decreased from 0.02404 to 0.0207 min^−1^ as the pH increased from 3.0 to 6.0, and then decreased to 0.01909 and 0.01769 min^−1^ when the pH was adjusted to 9.0 and 12.0, respectively (Appendix A). This phenomenon can be explained by the surface charge status of the catalyst and the existing PMS forms at different pH values. The pH_pzc_ of Co-Ni-Mo/CNT was measured to be 5.63 (Appendix A), making Co-Ni-Mo/CNT positively charged in acidic solutions. The pKa2 value of PMS is approximately 9.4 [50]. This means HSO_5_^−^ was the dominant species whether the initial solution was acidic, neutral, or slightly alkaline. Thus, the electrostatic interaction between Co-Ni-Mo/CNT and HSO_5_^−^ in the acidic solution was favorable during the activation process, promoting enhanced degradation. In contrast, electrostatic repulsion in neutral and alkaline environments inhibits PMS activation. In addition, under strong alkalinity, PMS became less active, as shown in Equations (1) and (2) [51], which significantly slowed the degradation rate compared to the acidic solution.
HSO_5_^−^ + ^•^OH → SO_5_^•−^ + H_2_O(1)
HSO_5_^−^ + SO_4_^•−^ → SO_5_^•−^ + SO_4_^2−^ + H^+^
(2)

The dosage of PMS has a direct influence on the decolorization of Rhodamine B. To investigate this effect, the impact of PMS dosage ranging from 0.5 to 4 g/L was evaluated. As shown in Figure 5b, the degradation rate of Rhodamine B increased with increasing PMS dosage. When the PMS dosage was 2 g/L, the degradation of Rhodamine B showed the best removal effect, with a degradation efficiency exceeding 99% at 60 min and a reaction rate of 0.8929 min^−1^ (Appendix A). This can be attributed to the fact that the amount of reactive oxygen species (ROS) was affected by the PMS concentration. As the amount of PMS increased, the solution exhibited a higher concentration of ROS, resulting in accelerated oxidation rates.

The influence of the catalyst dosage is shown in Figure 5c. The degradation rate of Rhodamine B exhibited an increase as a function of the catalyst dosage. Specifically, when the catalyst dosage was increased from 0 to 0.2 g/L, the degradation rate improved from 0.01012 min^−1^ to 0.0917 min^−1^ (Appendix A). This increase in degradation efficiency can be attributed to the activation of PMS into potent ROS, such as SO_4_^•−^, ^•^OH, and ^1^O_2_, which can be facilitated by the presence of a catalyst. Consequently, the breakdown of pollutants was accelerated when more catalysts were added to the system. Of particular significance was the observation that when the catalyst input was as small as 0.1 g/L, the degradation performance was comparable with 0.2 g/L dosage, with over 99% degradation of Rhodamine B achieved within a 60 min timeframe. This research offers exciting opportunities to achieve rapid degradation with such a small dosage of catalyst, which would greatly increase the economic feasibility of the application.

The effect of the initial Rhodamine B concentration on the degradation was further examined (Figure 5d). The degradation rate decreases as the initial dye concentration increases. At the initial concentration of 200 mg/L, 92% degradation was achieved within 60 min. As the concentrations in real dye wastewater are usually high, Co-Ni-Mo/CNT exhibits great potential as a catalyst for more efficient wastewater treatment processes.

The experimental findings depicted in Figure 5e reveal that the Rhodamine B degradation efficiency is influenced by the introduction of different anions. The molar quantity of added anions was kept constant, and it was observed that among the tested anions, the addition of Cl^−^ had little impact on the degradation rate, as the degradation efficiency surpassed 99% at 60 min. Conversely, the presence of PO_4_^3−^, CO_3_^2−^, and SO_4_^2−^ exerted different amounts of influence on the degradation rate. Inorganic anions, such as SO_4_^2−^, CO_3_^2−^, and PO_4_^3−^ have been reported to compete with ROS at adsorption sites or quench the generated free radicals [52,53,54], resulting in inhibition of the degradation. PO_4_^3−^ significantly inhibited the reaction, with a degradation rate of 20.2% in 60 min. The influence of common metal cations in groundwater was also examined (Figure 5f). The presence of Ca^2+^ and Mg^2+^ had minimal impact on the degradation efficiency of Rhodamine B, indicating that these cations do not significantly interfere with the catalytic performance of Co-Ni-Mo/CNT in the PMS activation process.

The findings depicted in Figure 6a suggest that the degradation performance enhanced as the reaction temperature increased from 293 K to 313 K, and the degradation rate increased from 0.06393 min^−1^ to 0.09746 min^−1^ (Appendix A). The reaction rate constant and reaction temperature fit the Arrhenius equation well (Figure 6b). The apparent activation energies were estimated to be 44.01 kJ/mol. This value is lower than that of the Co_x_Fe_3−x_O_4_ composite (49.01 kJ/mol) [55], Fe_3_O_4_/Co_3_S_4_ nanosheets (48.85 kJ/mol) [56], and MoS_2_ (59.05 kJ/mol) [23], indicating that Co-Ni-Mo/CNT has high catalytic activity.

To further elucidate the synergy of the ternary metal sulfides, an electrochemical evaluation was conducted. Appendix A shows the impedance spectroscopy (EIS) spectra of the Co-Ni-Mo/CNT, Co-Ni-Mo/CNT + PMS, and Co-Ni-Mo/CNT + PMS + Rhodamine B systems. Each spectrum shows a semicircle and straight line representing the charge transfer and diffusion processes, respectively. When both PMS and Rhodamine B were added, the equivalent radius of the semicircle increased significantly compared to that of Co-Ni-Mo/CNT and Co-Ni-Mo/CNT + PMS, indicating a dramatic charge transfer process between the catalyst, PMS, and the reactant [57]. The EIS profiles of Co-Ni-Mo/CNT, Co-Mo/CNT, and Ni-Mo/CNT in the presence of PMS are shown in Appendix A. Co-Ni-Mo/CNT exhibited a smaller charge transfer resistance than Co-Mo/CNT and Ni-Mo/CNT, implying that the reaction was favorable on the surface of Co-Ni-Mo/CNT; thus, a faster degradation process was achieved. Cyclic voltammetry (CV) curves of different catalysts with PMS in a Na_2_SO_4_ electrolyte were used to observe the electron transfer process at the water-catalyst interface, as shown in Appendix A. All the electrodes showed typical oxidation and reduction peaks of faradaic pseudocapacitance, which indicated a redox reaction between the catalysts and PMS [58]. Co-Ni-Mo/CNT exhibits higher current densities at the anodic and cathodic peaks than the other catalysts, which is consistent with its higher conductivity. It has been reported that multi-metal doping can optimize the electronic structure of metal sulfides, thereby reducing the energy barrier of the reaction [59]. This explains the outstanding degradation performance of Co-Ni-Mo/CNT, in which the synergistic effect between Mo and Co-Ni sulfide dramatically reduces the reaction barrier.

### 2.4. Identification of ROS in Co-Ni-Mo/CNT System

To explore the role of ROS in the degradation process, the performance of Co-Ni-Mo/CNT in the presence of different scavengers was investigated (Figure 7a). Methyl alcohol can capture SO_4_^•−^ and ^•^OH at comparable rates [60], whereas tertiary butanol (TBA) can capture SO_4_^•−^ slower than ^•^OH [55]. Thus, methanol and TBA are widely used to identify SO_4_^•−^ and ^•^OH radical species. As shown in Figure 7a, no retardation of Rhodamine B degradation was observed when methanol or TBA was used as scavengers in the system. Considering the hydrophobic status of the Co-Ni-Mo/CNT surface, which is attributed to the hydrophobic nature of MoS_2_ and CNT [61,62,63], it may be difficult to quench the active radicals bound to the Co-Ni-Mo/CNT surface using hydrophilic alcohols [64,65]. Therefore, dimethyl sulfoxide (DMSO) was used instead of alcohols to detect surface-bound radicals. Rhodamine B degradation was slightly inhibited in the presence of DMSO, illustrating the minor contributions of surface-bound SO_4_^•−^ and ^•^OH. Furthermore, when L-histidine, a scavenger of ^1^O_2_ [66], was applied, a significant deceleration of the degradation was observed, indicating the key role of ^1^O_2_.

To further confirm the presence of ROS during degradation, EPR analysis was conducted using TEMP and DMPO as trapping agents, and the results are shown in Figure 7. The intensified triplet signal of TEMP-^1^O_2_ (1:1:1), shown in Figure 7b, confirms the generation of ^1^O_2_. Notably, the typical signals of DMPO-^•^OH (1:2:2:1) and DMPO-SO_4_^•−^ (1:1:1:1:1:1) in Figure 7c exhibited increased peak intensities at 0 and 5 min, suggesting the participation of ^•^OH and SO_4_^•−^ in the degradation process. In addition, the relatively weaker intensity of DMPO-SO_4_^•−^ peaks than that of DMPO-^•^OH peaks can be attributed to the rapid conversion from DMPO-SO_4_^•−^ to DMPO-^•^OH through nucleophilic substitution. Based on these findings, it can be concluded that the principal ROS involved in the degradation of Rhodamine B within this system is ^1^O_2_, followed by surface-bound ^•^OH and SO_4_^•−^.

### 2.5. Degradation Pathway of Rhodamine B

To determine the degradation pathway of Rhodamine B in the Co-Ni-Mo/CNT + PMS system, UPLC-MS was employed to analyze the intermediates in the solution. The UPLC spectra of Rhodamine B solution with Co-Ni-Mo/CNT + PMS at various times are shown in Appendix A. The peak absorbance of Rhodamine B at 8.66 min decreased gradually and eventually disappeared after 40 min, indicating the complete destruction of the chromophore structure. The intermediates formed during degradation were analyzed by MS and are summarized in Appendix A. Based on the intermediates formed during the degradation process, two possible degradation pathways are proposed, as shown in Figure 8. First, Rhodamine B undergoes hydroxylation by ^•^OH to form a hydroxylation intermediate (*m*/*z* = 459.2), which further degrades into organic acids and alcohols [67]. Another method involves deethylation, decarboxylation, deamination, cleavage, ring opening, and mineralization. Firstly, Rhodamine B is deethylated gradually to form C_26_H_27_N_2_O_3_^+^ (*m*/*z* = 415.2) and C_24_H_23_N_2_O_3_^+^ (*m*/*z* = 387.2) by the continuous attack of ^•^OH SO_4_^•−^ and ^1^O_2_. Second, C_19_H_18_N_2_O (*m*/*z* = 290.1) is formed after N-deethylation and decarboxylation reactions. The intermediates are further deaminated to form C_19_H_14_O (*m*/*z* = 258.1), which is followed by cleavage and ring-opening reactions to generate C_18_H_14_O (*m*/*z* = 246) and C_17_H_18_O (*m*/*z* = 238) [68]. Finally, the intermediates decompose into organic matter such as alcohols, amines, and organic acids with low molecular weight and low toxicity and finally mineralize to CO_2_ and H_2_O [69].

Based on the aforementioned analysis, a plausible mechanism for the PMS activation on Co-Ni-Mo/CNT can be expressed by Equations (3)–(18). Firstly, Co^2+^ and Ni^2+^ react with PMS to form SO_4_^•−^ (Equation (3)) [70]. Meanwhile, because of the reducing ability of Mo^4+^, a redox reaction can also occur between the MoS_2_ and PMS molecules (Equations (6)–(9)), generating SO_4_^•−^ and ^•^OH [22,71,72]. The recoveries of Co^2+^ and Ni^2+^ are achieved via two pathways. One path is the reduction of Co^3+^/Ni^3+^ by PMS, as shown in Equation (4). The other pathway involves the reaction between Co^3+^/Ni^3+^ and MoS_2_ according to Equation (5) [32,73]. Therefore, the regeneration of Co^2+^/Ni^2+^ is enhanced, guaranteeing a continuous catalytic reaction. Furthermore, the oxidized CNTs can activate PMS by generating sulfate radicals (Equation (10)) [74]. The introduction of CNTs also contributes to the enhancement of active sites and the electron transfer ability between metal sulfides and PMS, thus ensuring high activation efficiency. The resultant SO_4_^•−^ subsequently reacts with water molecules to produce ^•^OH (Equation (11)). Simultaneously, PMS undergoes hydrolysis, yielding O_2_^•−^ (Equations (12)–(14)) [75] and facilitating the decomposition of PMS into SO_4_^•−^ (Equation (15)). Within this oxidation system, both self-activation and self-decomposition of O_2_^•−^ give rise to ^1^O_2_ (Equation (17)) [73], whereas the interactions of O_2_^•−^and SO_5_^•−^ with water molecules lead to the formation of ^1^O_2_ (Equation (18)) [76]. Moreover, Mo^6+^ can participate in the reaction with O_2_^•−^ to produce ^1^O_2_ (Equation (16)) [77]. Ultimately, Rhodamine B is attacked by ROS to form small molecules. The degradation mechanism is illustrated in Figure 9.
Co^2+^/Ni^2+^ + HSO_5_^−^ → Co^3+^/Ni^3+^ + SO_4_^•−^ +H_2_O(3)
Co^3+^/Ni^3+^ + HSO_5_^−^ → Co^2+^/Ni^2+^ + SO_5_^•−^ + H^+^(4)
Co^3+^/Ni^3+^ + Mo^4+^ → Co^2+^/Ni^2+^ + Mo^5+^/Mo^6+^(5)
Mo^4+^ + HSO_5_^−^ → Mo^5+^ + SO_4_^•−^ + OH^−^(6)
Mo^4+^ + HSO_5_^−^ → Mo^5+^+ SO_4_^2−^ + ^•^OH(7)
Mo^5+^ + HSO_5_^−^ → Mo^6+^+ SO_4_^•−^ + OH^−^(8)
Mo^5+^+ HSO_5_^−^ → Mo^6+^+ SO_4_^2−^ + ^•^OH(9)
CNT-C=O + HSO_5_^−^ → CNT-C-O* + SO_4_^•−^ + OH^−^(10)
SO_4_^•−^ + H_2_O/OH^−^ → HSO_4_^−^/SO_4_^2−^+ ^•^OH(11)
HSO_5_^−^ → SO_5_^2−^ + H^+^(12)
SO_5_^2−^ + H_2_O → O_2_^•−^ + SO_4_^2−^ + H^+^(13)
HSO_5_^−^ + O_2_^•−^ → SO_4_^•−^ + O_2_ + OH^−^(14)
HSO_5_^−^ → SO_4_^•−^ + ^•^OH(15)
Mo^6+^+ O_2_^•−^ → Mo^6+^+ ^1^O_2_(16)
2O_2_^•−^ + 2H^+^ → ^1^O_2_ + H_2_O_2_(17)
HSO_5_^−^ + SO_5_^2−^ → HSO_4_^−^ + SO_4_^2−^ + ^1^O_2_(18)

### 2.6. Effect of Water Quality Matrix and Evaluation of Reusability

The effects of the water quality matrix were also investigated (Figure 10a). Degradation of the Rhodamine B solution was accomplished within approximately 40 min when the solution was diluted with deionized water. Conversely, when tap water was used, the degradation process of the Rhodamine B solution remained incomplete even after 60 min, with a total degradation rate of only 83%. This may be due to the presence of natural organic matter, dissolved organic matter, and other impurities in the water sample, which were adsorbed on the surface of the catalyst and prevented contact between Co-Ni-Mo/CNT and PMS [78]. The reusability results are shown in Figure 10b. In each cycle, the degradation efficiency reached 99% after 60 min, and the apparent degradation rate did not decrease, indicating that the activity of Co-Ni-Mo/CNT was maintained after four successive runs. This shows that the solid catalyst can be recycled by a simple washing process after degradation and has a long lifespan and good stability.

XRD image of the recycled catalyst is shown in Appendix A. The XRD results illustrate that all the peaks of the recycled catalyst remained the same as those of the fresh catalyst, except that the intensities of the peaks for NiCo_2_O_4_ decreased slightly owing to the dissolution of the metal element. These findings suggested that the catalyst structure was stable in the degradation system.

XPS was used to examine the stability and valence changes of the used catalyst (Figure 11). Although the peaks in the Co 2p and Ni 2p spectra after degradation are fuzzy, both spectra can be deconvoluted into two main peaks and two corresponding satellites. The S 2p spectra before and after degradation were similar, indicating the stable structure of the sulfides. In the Mo 3d spectrum, the peaks of Mo^4+^, Mo^5+^, and Mo^6+^ remained, while those of Mo^4+^ decreased from 50.3% to 23.6%, and those of Mo^5+^ and Mo^6+^ increased from 22.0% and 27.7% to 38.7% and 37.7%, respectively. This phenomenon implies that the Mo^4+^ on the surface of MoS_2_ was oxidized to Mo^5+^ and further oxidized to Mo^6+^ during the reduction of Co^3+^ and Ni^3+^ (Equation (5)) and PMS activation (Equations (6)–(9)), which can accelerate the degradation rate and maintain the continuous occurrence of the degradation reaction.

## 3. Materials and Methods

### 3.1. Chemicals

Rhodamine B (C_28_H_31_ClN_2_O_3_) and potassium peroxomonosulfate (KHSO_5_) were purchased from Shanghai Aladdin Biochemical Technology Co., Ltd, Shanghai, China. Methyl alcohol (CH_3_OH), Cobalt chloride hexahydrate (CoCl_2_·6H_2_O) and sodium phosphate tribasic (Na_3_PO_4_) were purchased from Yonghua Chemical Co., Ltd. (Suzhou, China). Carbon nanotube (CNT) was purchased from Chengdu Organic Institute (Chengdu, China). Thiourea (CN_2_H_4_S) and sodium molybdate dihydrate (Na_2_MoO_4_·2H_2_O) were purchased from Sahn Chemical Technology Co., Ltd. (Shanghai, China). Nickel chloride hexahydrate (NiCl_2_·6H_2_O) was purchased from Shanghai Macklin Biochemical Co., Ltd. (Shanghai, China). Sodium sulfate (Na_2_SO_4_) was purchased from Jiangsu Qiangsheng Functional Chemical Co., Ltd. (Changshu, China). Sodium carbonate (Na_2_CO_3_) was purchased from Shanghai Hongguang Chemical Factory (Shanghai, China). Sodium chloride (NaCl) was purchased from the Guangdong Chemical Reagent Engineering Technology Research and Development Center (Zhaoqing, China).

### 3.2. Preparation of Composite Materials

To prepare the Co-Ni-Mo/CNT catalyst, 80 mg of CNTs and 60 mL of distilled water were placed in a 100 mL black lining, and ultrasound was applied for 15 min. Then, four substances, including 0.0797 g CoCl_2_·6H_2_O, 0.4567 g CN_2_H_4_S, 0.0796 g NiCl_2_·6H_2_O, and 0.0811 g Na_2_MoO_4_· 2H_2_O, were added into the carbon nanotube dispersions, stirred magnetically for 15 min, and then treated with ultrasound for 1 h. The solution was then placed in the inner lining of a hydrothermal kettle and heated in an oven at 200 °C for 12 h. Finally, the black precipitate was washed with distilled water and ethanol thrice, and the product was dried at 60 °C for 8 h to obtain the as-prepared sample and was denoted as Co-Ni-Mo/CNT. The Co-Mo/CNT, Co/CNT, Ni/CNT, Mo/CNT, and Ni-Mo/CNT catalysts were prepared from the corresponding substances.

### 3.3. Characterization of Composite Materials

The phase structure was analyzed using XRD on an X-ray diffractometer (BrucerD8) with Cu-Kα radiation. The morphology was observed using a Hitachi S-4800 scanning electron microscope and a transition electron microscope (FEI Talos F200S). XPS was conducted with a Thermo Scientific K-Alpha+ instrument using an A1Kα X-ray source. The CNT content of the samples was estimated via thermogravimetric analysis using a thermal analyzer (STA 449 F3) at a heating rate of 10 °C/min under an air atmosphere. The surface areas of the samples were measured using the Brunauer–Emmett–Teller (BET) method on a MicroActive for ASAP 2460 (Version 2.02). N_2_ adsorption–desorption isotherm was recorded at 77 K, the specific surface area was calculated in the relative pressure range of 0.05–0.25, and the pore size distribution was evaluated by using the Barrette–Joynere–Halenda (BJH) model. Ion leaching concentrations during the degradation process were determined by inductively coupled plasma mass spectrometry. The radicals in the system were detected using EPR with a Bruker EMXplus instrument. A solution containing 0.1 g/L of the catalyst was stirred continuously, followed by the addition of PMS (0.5 g). The time was recorded as *t*_0_. At specific intervals, 0.5 mL of the solution was withdrawn and filtered through a polytetrafluoroethylene syringe filter disk into a vial. Then, 5,5-dimethyl-1-pyrroline N-oxide (DMPO, 100 mM) was added to capture SO_4_^•−^ and ^•^OH, while 2,2,6,6-tetramethyl-4-piperidine (TEMP,100 mM) was used to capture ^1^O_2_. The mixed liquid was immediately transferred into a quartz tube and inserted into the EPR sample chamber. Electrochemical measurements, such as CV and EIS, were performed in a three-electrode system on an electrochemical workstation (Iviumstat) containing 0.5 mM Na_2_SO_4_ as the electrolyte, and the prepared electrode, Pt electrode, and Ag/AgCl electrode were selected as the working, counter, and reference electrodes, respectively. The working electrode was fabricated by coating a mixture of catalyst powder, carbon black, and polyvinylidene fluoride onto a 1 × 1 cm^2^ Ti foil. The intermediates formed during the degradation process were analyzed by electrospray ionization mass spectrometry (ESI-MS) using a Waters Acquity UPLC h-class instrument equipped with a C18 column. The mobile phase consisted of 0.1% formic acid/water (A) and acetonitrile (B), at a flow rate of 0.4 mL/min. At specified intervals, the dye solution was filtered through a Minisart RC filter and injected into the UPLC column. The detection wavelength was set at 554 nm.

### 3.4. Degradation Process

The Co-Ni-Mo/CNT catalyst was introduced into the Rhodamine B solution and stirred continuously until adsorption equilibrium was reached. Subsequently, PMS was added under vigorous stirring, and the reaction time was denoted as the beginning of the reaction. Subsequently, 2 mL of the sample was drawn at regular intervals and immediately added to 2 mL of methanol to terminate the reaction. The mixture was then filtered through a polytetrafluoroethylene syringe filter disc and the concentration was measured at a wavelength of 554 nm using an Ultraviolet–Visible Spectrophotometer (UV-5500PC). Each experiment was performed in duplicate under the same conditions, and the degradation curves are presented as averages and error bars. The experiments were conducted twice under various conditions to determine the optimal conditions. For reusability, Co-Ni-Mo/CNT was separated from the decomposed liquor using a centrifuge and washed thoroughly with ethanol and deionized water to remove the absorbed dye and PMS. After drying completely at 60 °C, Co-Ni-Mo/CNT was placed in a fresh Rhodamine B solution for the next degradation cycle.

## 4. Conclusions

The catalyst was constructed by combining ternary metal sulfides with CNT, and its application in PMS activation and Rhodamine B degradation was investigated. The Co-Ni-Mo/CNT catalyst demonstrated outstanding activation properties compared to those of Co-Mo/CNT, Ni-Mo/CNT, and Mo/CNT. This catalyst also exhibited exceptional adaptability and high degradation efficiency for various organic contaminants, including bisphenol A, berberine hydrochloride, methyl orange, Congo red, and phenol. Several factors, including the input of potassium persulfate, catalyst dosage, the initial concentration of the Rhodamine B solution, temperature, pH, the type of ion input, and the water quality matrix, were investigated to determine the optimal degradation conditions. The results revealed that at an initial concentration of 100 mg/L of Rhodamine B, with an input of 2 g/L of potassium persulfate, 0.1 g/L of catalyst dosage, and a temperature of 40 °C, the degradation rate exceeded 99% within a remarkably short time of 20 min. Moreover, the Co-Ni-Mo/CNT + PMS system exhibited high degradation efficiency after four cycles, implying good stability and reusability. Through radical quenching experiments and EPR analysis, it was determined that the primary active group in the Co-Ni-Mo/CNT + PMS system was ^1^O_2_, SO_4_^•−^ and ^•^OH. Our results provide a new and effective strategy for designing multicomponent composites for advanced oxidation processes. 

## Figures and Tables

**Figure 1 molecules-29-03633-f001:**
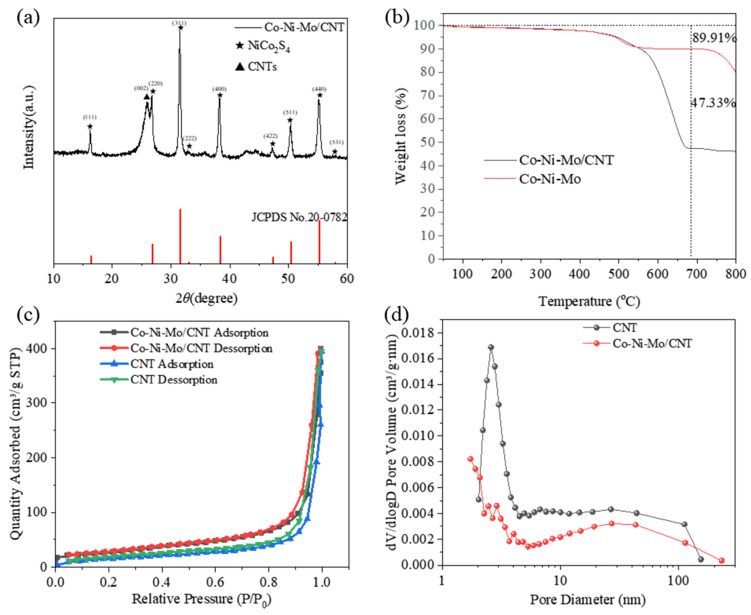
(**a**) XRD patterns of Co-Ni-Mo/CNT; (**b**) TGA curves of Co-Ni-Mo/CNT and Co-Ni-Mo; (**c**) N_2_ adsorption–desorption isotherms; and (**d**) pore size distribution of Co-Ni-Mo/CNT and CNT.

**Figure 2 molecules-29-03633-f002:**
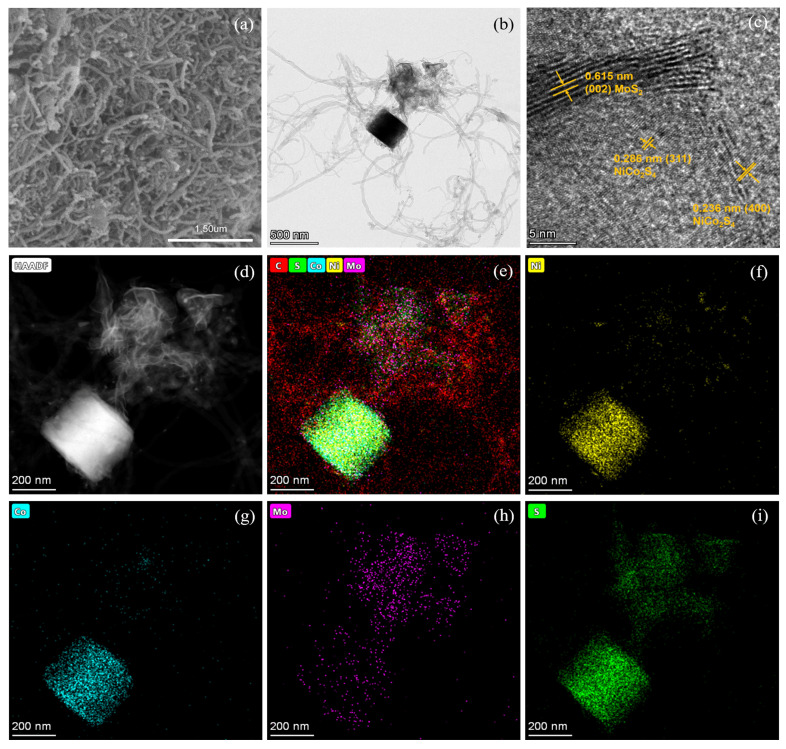
(**a**) SEM; (**b**) TEM; (**c**) high-resolution TEM; and (**d**–**i**) EDX mapping images of Co-Ni-Mo/CNT.

**Figure 3 molecules-29-03633-f003:**
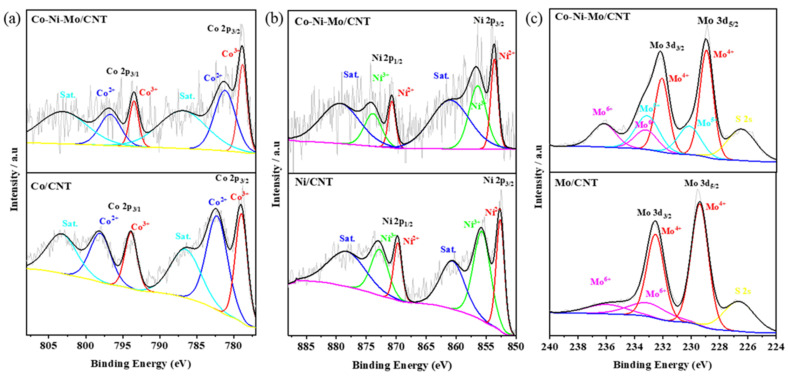
XPS patterns of (**a**) Co 2p; (**b**) Ni 2p; and (**c**) Mo 3d of Co-Ni-Mo/CNT, Co/CNT, Ni/CNT, and Mo/CNT.

**Figure 4 molecules-29-03633-f004:**
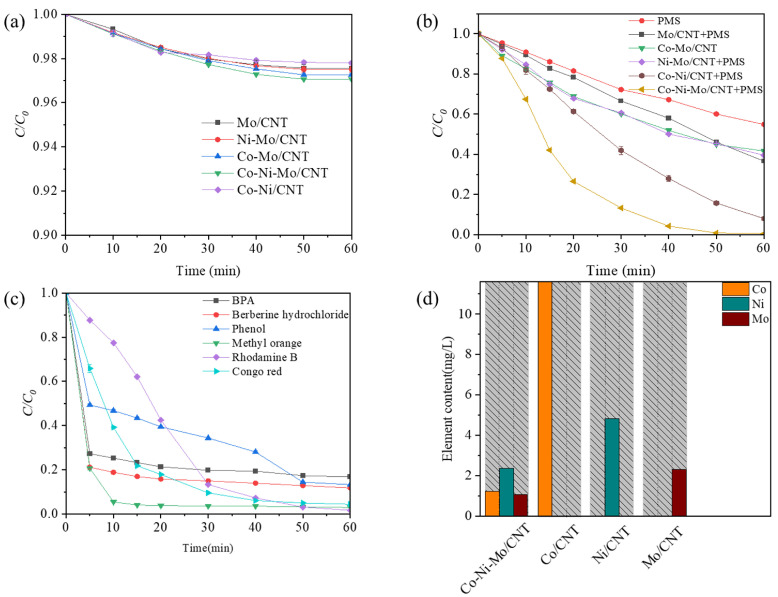
(**a**) Adsorption of Rhodamine B in different systems; (**b**) degradation of Rhodamine B in different systems; (**c**) degradation of different pollutants by Co-Ni-Mo/CNT/PMS system; and (**d**) leaching ion concentrations in different systems (reaction conditions: *C*_0_ = 0.1 g/L, *C*_catalyst_ = 0.1 g/L, *C*_PMS_ = 2 g/L, *T* = 20 °C, and natural pH).

**Figure 5 molecules-29-03633-f005:**
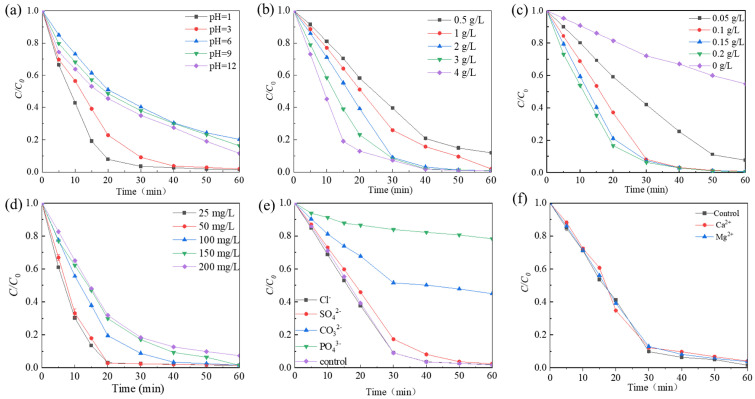
The effects of (**a**) different pH; (**b**) PMS concentration; (**c**) catalyst dosage; (**d**) Rhodamine B concentration; (**e**) various anion species; and (**f**) various cation species.

**Figure 6 molecules-29-03633-f006:**
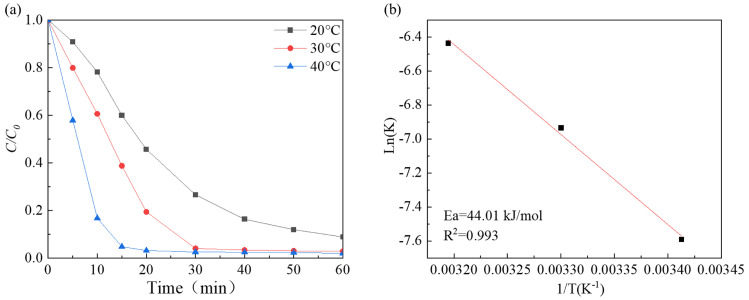
(**a**) The effects of different temperatures; (**b**) the fitting of Arrhenius equation.

**Figure 7 molecules-29-03633-f007:**
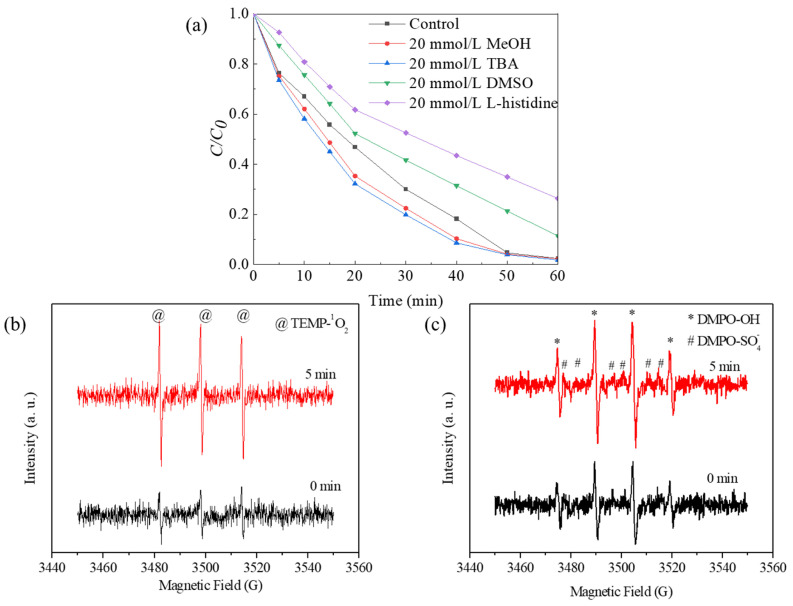
(**a**) Degradation effects of different scavengers on Rhodamine B; EPR spectra in presence of (**b**) TEMP and (**c**) DMPO in Co-Ni-Mo/CNT catalytic reaction system (reaction conditions: *C*_catalyst_ = 0.1 g/L, *C*_PMS_ = 2 g/L, *T* = 20 °C, and natural pH).

**Figure 8 molecules-29-03633-f008:**
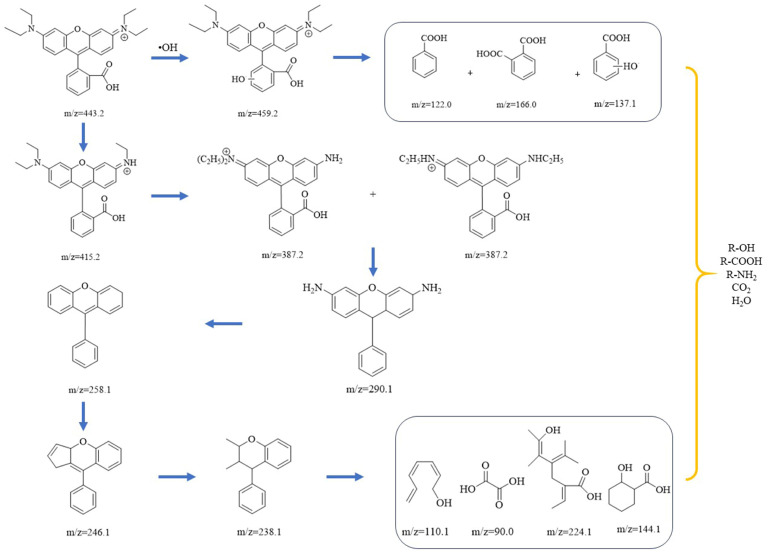
Possible degradation pathways of Rhodamine B in the Co-Ni-Mo/CNT+PMS system.

**Figure 9 molecules-29-03633-f009:**
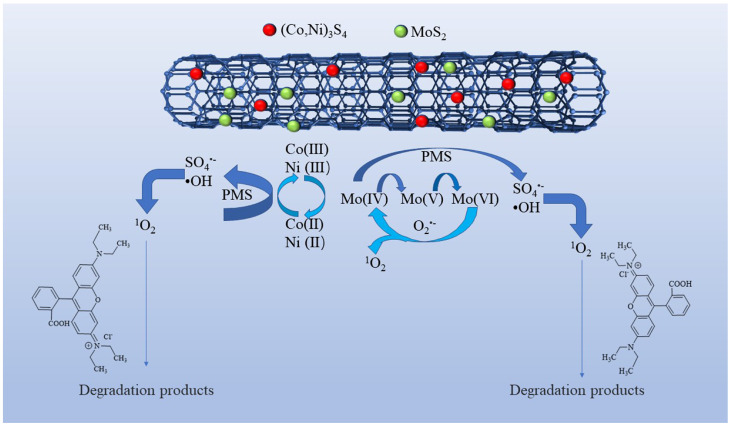
PMS activation mechanism and Rhodamine B degradation on Co-Ni-Mo/CNT.

**Figure 10 molecules-29-03633-f010:**
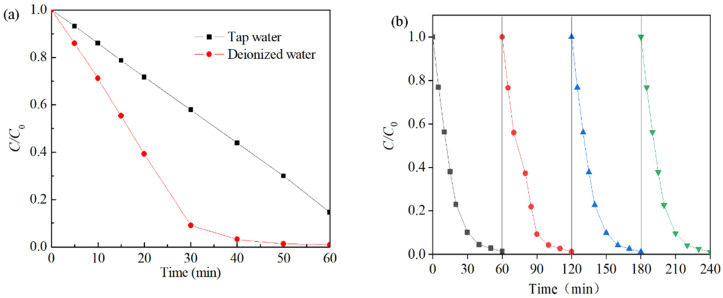
(**a**) Effects of different water quality substrates and (**b**) reusability tests of Co-Ni-Mo/CNT (reaction conditions: *C*_0_ = 0.1 g/L, *C*_catalyst_ = 0.1 g/L, *C*_PMS_ = 2 g/L, *T* = 20 °C, and natural pH).

**Figure 11 molecules-29-03633-f011:**
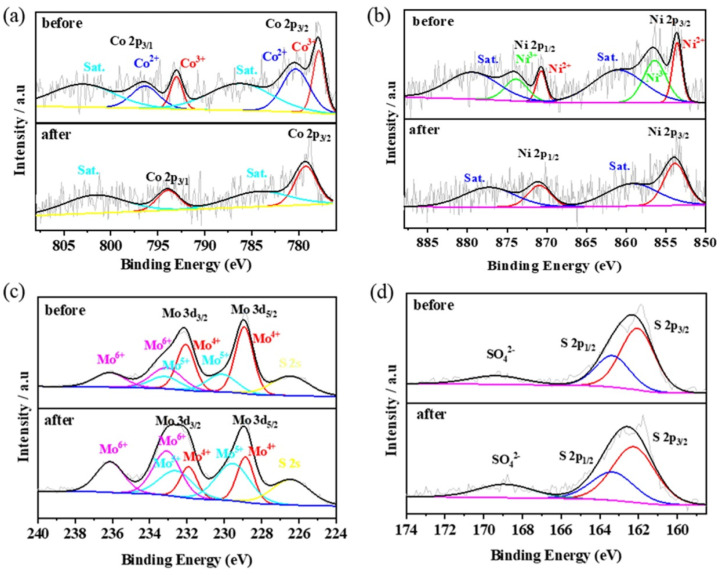
XPS patterns of Co 2p (**a**), Ni 2p (**b**), Mo 3d (**c**), and S 2p (**d**) of Co-Ni-Mo/CNT before and after degradation.

**Table 1 molecules-29-03633-t001:** Comparison of degradation performances of different catalysts.

Catalyst	PMS Concentration	Dye Concentration	Catalyst Dosages	Degradation Rate	Reference
CoFe_2_O_4_/OMC	1.5 mM	100 mg/L	0.05 g/L	Almost complete degradation @ 100 min	[5]
CoMgFe-LDH	1.2 mM	90 μM	0.8 g/L	94.3% @ 10 min	[46]
CoFe@CCB	0.3 mM	50 mg/L	0.04 g/L	99% @ 15 min	[47]
Co-NC-x	0.025 mM	80 mg/L	0.8 g/L	25% @ 8 min	[48]
MoS_2_/Co_0.75_ Mo_3_S_3.75_	0.06 mM	50 mg/L	0.3 g/L	21% @ 98 min	[31]
Co_3_O_4_-Fe_3_O_4_	0.5 mM	180 mg/L	0.1 g/L	98% @ 15 min	[49]
Co-Ni-Mo/CNT	2g/L	100 mg/L	0.1 g/L	Almost complete degradation @ 20 min	This study

## Data Availability

The original contributions presented in the study are included in the article (and Appendix A), further inquiries can be directed to the corresponding authors.

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
