# Peer review of "Activation of Peroxymonosulfate by Co-Ni-Mo Sulfides/CNT for Organic Pollutant Degradation"

_molecules, 2024, doi:10.3390/molecules29153633_

Round 1

Reviewer 1 Report

Comments and Suggestions for Authors

Manuscript is interesting and suitable for publication in Molecules. Authors should address following observations.

1. Some figures are blur, their resolution should be improved.

2. Degradation mechanism should be explained in detail.

3. Effect of different cations and anions on degradation should also be studied.

Author Response

Comment 1: Some numbers are blur, the resolution should be increased.

Response 1: Thanks for the suggestion. We've improved the resolution of all numbers.

Comment 2: The degradation mechanism should be explained in detail.

Response 2: Thank you for your suggestion. We have explored the degradation pathway by analyzing the intermediates using UPLC-MS in the revised manuscript. Based on the intermediates formed during the degradation process, two possible degradation pathways are proposed, as shown in Fig. 8. First, Rhodamine B undergoes hydroxylation by OH to form a hydroxylation intermediate (m/z=459.2), which further degrades into organic acids and alcohols. Another method involves deethylation, decarboxylation, deamination, cleavage, ring opening, and mineralization. Firstly, Rhodamine B is deethylated gradually to form C26H27N2O3+ (m/z=415.2) and C24H23N2O3+ (m/z=387.2) by the continuous attack of OH SO4•- and 1O2. Second, C19H18N2O (m/z=290.1) is formed after N-deethylation and decarboxylation reactions. The intermediates are further deaminated to form C19H14O (m/z=258.1), which is followed by cleavage and ring-opening reactions to generate C18H14O (m/z=246) and C17H18O (m/z=238). Finally, the intermediates decompose into organic matter such as alcohols, amines, and organic acids with low molecular weight and low toxicity, and finally mineralize to CO2 and H2O.

Comment 3: The effects of different cations and anions on degradation should also be studied.

Response 3:  Thanks for your suggestion. In the previous manuscript, we have studied the effects of Cl, SO42-, CO32-, and PO43- on degradation. The experimental findings depicted in Fig. 5(e) reveal that the degradation efficiency of Rhodamine B is influenced by the introduction of different anions. The addition of Cl had little impact on the degradation rate, as the degradation efficiency surpassed 99% at 60 min. Conversely, the presence of PO43-, CO32-, and SO42- exerted different amount of influence on the degradation rate. Among them, PO43- significantly inhibited the reaction with a degradation of 20.2% in 60 min. In the revised manuscript, we studied the influence of most common metal cations such as Ca2+ and Mg2+ in groundwater (Fig. 5(f)). It shows that the presence of Ca2+ and Mg2+ had little effect on the degradation.

Reviewer 2 Report

Comments and Suggestions for Authors

The manuscript is clear and presented in a well-structured manner. The experimental was designed appropriately. The discussion of the results is justified. However, I have few comments as following:

  • According to the result in Fig 4b, authors should discuss the reasons why Co-Ni-Mo/CNT could catalyze the degradation of methyl orange mostly, followed by berberine hydrochloride, BPA, and phenol. 
  • Authors should include details regarding the statistical analysis such as error bars in the figures. 

Author Response

Comment 1: Based on the results of Figure 4b, the authors should discuss the reasons why Co-Ni-Mo/CNT mainly catalyzes the degradation of methyl oranges, followed by berberine hydrochloride, BPA, and phenol.

Reply 1: Thank you for your question. In our study, methyl orange (MO) was degraded better than berberine hydrochloride, BPA, and phenol. Other researchers observed a similar phenomenon, with MO degrading more quickly than berberine hydrochloride and phenol (Y. Zhang et al., Journal of Chemical Engineering 373 (2019) 179-191; W. Ma et al., Chemical Engineering Journal 336 (2018) 721-731). The reason may be related to the electrostatic interaction between the containment vessel and the catalyst. According to our study, the catalyst is positively charged in the degradation system (pH<5, Section 2.2). MO is an anionic dye, whereas berberine hydrochloride is positively charged in solution. When the pH value of the solution is lower than the pKa value of BPA (8.0) and phenol (10.0), BPA and phenol mainly exist in the form of neutral molecules. As a result, the electrostatic interaction between MO and the catalyst is strongest among these containments, which significantly enhances degradation.

Comment 2: The author should include details about the statistical analysis, such as error bars in the plot.

Response 2: Thank you for checking it carefully. In fact, the experiments were performed in duplicate or triplets, and error bars were shown in all degradation plots. Some errors are too small to be identified in the numbers. For example, in Figure 4(b), all error bars are shown, but only a few are visible.

Reviewer 3 Report

Comments and Suggestions for Authors

The authors reported organic pollutant degradation via the activation of peroxymonosulfate by Co-Ni-Mo sulfides/CNT. The submission could be accepted after revision taking into account the following points:-

1.       Figure 1a for XRD shows NiCo2S4 (JCPDS No. 20-0782). Where can we see the other phases of Co-Ni-Mo?

2.       An adsorption profile should be included showing the adsorption maximum capacity.

3.       Dye selectivity should be investigated using different anionic and cationic dyes.

4.       Catalysts after recyclability should be fully characterized.

5.       Mass spectrometry should be added for Rhodamine B after catalysis confirming the degradation and showing the degradation products.

6.       Further analysis should be included to support the mechanism in Figure 7.

7.       A comparison with previously published catalysts should discussed and summarized in a Table.

8.       RhB was degraded using different catalysts. Thus, references should be updated suggesting e.g., https://doi.org/10.1002/EXP.20230050; DOI: 10.1039/D2RA00503D; https://doi.org/10.1016/j.jelechem.2024.118072; https://doi.org/10.1016/j.jece.2024.112547. The advantages and disadvantages of each material should be discussed.

9.       The language should be revised and typos should be corrected.

Minors

10.   Remove redundant words such as ‘novel’

11.   Decrease significant figure number for values such as ‘0.4566’; ‘0.6166’; ……

12.   ‘PH’ in Figure 5a, should be ‘pH’

Comments on the Quality of English Language

Professional language service should be conducted.

Author Response

Comments 1: Figure 1a for XRD shows NiCo2S4 (JCPDS No. 20-0782). Where can we see the other phases of Co-Ni-Mo?

Response 1: Thank you for your careful checking. There were no significant peaks of MoS2 in XRD, suggesting the amorphous nature or highly dispersed status of MoS2 in Co-Ni-Mo/CNT. But Mo element was detected by EDX in Fig. 2. Mo4+ species and Mo-S bonds were detected by XPS in Fig. 3 as well, proving the presence of MoS2 in the composite.

Comments 2: An adsorption profile should be included showing the adsorption maximum capacity.

Response 2: Thank you for your suggestion. The adsorption experiments of different catalysts have been done and the profiles were shown in Fig. 4(a). The adsorption percentages of Rhodamine B by Co-Ni-Mo/CNT, Co-Ni/CNT, Co-Mo/CNT, Ni-Mo/CNT, and Mo/CNT were 2.9%, 2.2%, 2.7%, 2.5%, and 2.4%, respectively, demonstrating the adsorption effect could be ignored.

Comments 3: Dye selectivity should be investigated using different anionic and cationic dyes.

Response 3:  Thanks for your suggestion. We studied the anionic dye such as Congo red (CR) and Methyl Orange (MO) in the revised manuscript (Fig. 4(c)). Anionic azo dyes exhibited notably higher degradation rates and efficiencies compared to cationic dyes like Rhodamine B. Specifically, MO was removed with approximately 99% efficiency within 10 minutes, while CR was degraded by 95.6% within 40 minutes. The enhanced performance for anionic azo dyes can be attributed to the favorable adsorption interactions between the contaminants and the catalyst. The initial pH values of the dye and PMS solutions were below 5 due to the hydrolysis of PMS, which makes the Co-Ni-Mo/CNT surface positively charged (pHpzc of 5.63, see Figure S2). This positively charged surface promotes the adsorption of anionic dyes, thereby improving degradation efficiency.

Comments 4: Catalysts after recyclability should be fully characterized.

Response 4:  We sincerely appreciate the valuable comments. We examined XRD and XPS of the recycled catalyst. XRD result illustrates that all the peaks of the recycled catalyst remained the same with the fresh catalyst, except that the intensities of peak for NiCo2O4 decreased slightly due to the dissolution of the metal element. The finding suggests that the catalyst structure is stable in the degradation system. XPS implied that Mo4+ on the surface of MoS2 was oxidized to Mo5+ and further to Mo6+ during the degradation, which can accelerate the degradation rate and keep the degradation reaction occurring continuously.

Comments 5:  Mass spectrometry should be added for Rhodamine B after catalysis confirming the degradation and showing the degradation products.

Response 5:   Thank you for your suggestion. We have explored the degradation pathway by analyzing the intermediates using UPLC-MS. Based on the intermediates formed during the degradation process, two possible degradation pathways are proposed, as shown in Fig. 8. First, Rhodamine B undergoes hydroxylation by OH to form a hydroxylation intermediate (m/z=459.2), which further degrades into organic acids and alcohols. Another method involves deethylation, decarboxylation, deamination, cleavage, ring opening, and mineralization. Firstly, Rhodamine B is deethylated gradually to form C26H27N2O3+ (m/z=415.2) and C24H23N2O3+ (m/z=387.2) by the continuous attack of OH SO4•- and 1O2. Second, C19H18N2O (m/z=290.1) is formed after N-deethylation and decarboxylation reactions. The intermediates are further deaminated to form C19H14O (m/z=258.1), which is followed by cleavage and ring-opening reactions to generate C18H14O (m/z=246) and C17H18O (m/z=238). Finally, the intermediates decompose into organic matter such as alcohols, amines, and organic acids with low molecular weight and low toxicity, and finally mineralize to CO2 and H2O.

Comments 6: Further analysis should be included to support the mechanism in Figure 7.

Response 6:  We performed XPS to examine the valence changes of the used catalyst (Fig. 11). In Mo 3d spectrum, the peaks of Mo4+, Mo5+, and Mo6+ remained, with the contents of Mo4+ decreased from 50.3% to 23.6%, while Mo5+ and Mo6+ increased from 22.0% and 27.7% to 38.7% and 37.7%, respectively. This phenomenon implied that Mo4+ on the surface of MoS2 was oxidized to Mo5+ and further to Mo6+ during the reduction of Co3+ and Ni3+ (Eq. 5) and the activation of PMS (Eq. 6-9), which can accelerate the degradation rate and keep the degradation reaction occurring continuously. Unfortunately, the peaks in Co 2p and Ni 2p spectra after the degradation were too fuzzy to calculate the ratio of Co2+/Co3+ and Ni2+/Ni3+. But the redox cycle of Co2+/Co3+ and Ni2+/Ni3+ during PMS activation are widely accepted in the former studies (J. Li, et al., J. Di, et al. Chemical Journal Engineering, 2023, 453, 139972; Journal of Water Process Engineering, 2020, 37, 101386; X. Dong et al., Applied Catalysis B: Environmental, 2020, 272, 118971; ).

Comments 7:  A comparison with previously published catalysts should discussed and summarized in a Table.

Response 7:  Thanks for your valuable suggestion. A literature research on the published catalysts were studied and summarized in Table 1 in the revised manuscript.

Comments 8: RhB was degraded using different catalysts. Thus, references should be updated suggesting e.g., https://doi.org/10.1002/EXP.20230050; DOI: 10.1039/D2RA00503D; https://doi.org/10.1016/j.jelechem.2024.118072; https://doi.org/10.1016/j.jece.2024.112547. The advantages and disadvantages of each material should be discussed.

Response 8:  Thanks for your valuable suggestion. These references are helpful to address the current development of catalysts for RhB degradation. All the references above were cited and the advantages and disadvantages have been discussed in the introduction part in the revised manuscript.

Commenrs 9:  The language should be revised and typos should be corrected.

Response 9:  Thank you for your suggestion. The manuscript has been edited by an English editing agency (www.editage.cn). Many grammatical or typographical errors have been revised. The modified parts were highlighted in the revised manuscript.

Comments 10: Remove redundant words such as ‘novel’.

Comments 11: Decrease significant figure number for values such as ‘0.4566’; ‘0.6166’; ……

Comments 12: ‘PH’ in Figure 5a, should be ‘pH’

Response 10-12:  Thanks for your careful checking. We have revised the manuscript and correct the above mistakes.

Reviewer 4 Report

Comments and Suggestions for Authors

This article is very interested, very well written, very well described and the system have been well characterized but I have some questions for the authors :

1. In Fig.7, the mechanism shows the total mineralization of Rhodamine B in CO2, H2O and other products. What are the nature of other products ? Are they toxic ? What are the technics used for the characterisation of the middle after the Rhodamine B degradation ?

2. Did you characterized the spent catalyst by EPR spectroscopy ? The EPR is very sensitive and useful for Mo5+, Ni2+, Co2+ species etc.....

3. Please give more informations about the EPR methodology for EPR experiments

4. Have you carried out a statistical analysis to determine the average length of the nano-sheets and their average stacking? Do you think that there is a relationship between this parameter and the degradation of Rhodamine B ?

Author Response

Comments 1: In Fig.7, the mechanism shows the total mineralization of Rhodamine B in CO2, H2O and other products. What are the nature of other products ? Are they toxic ? What are the technics used for the characterisation of the middle after the Rhodamine B degradation ?

Response 1:  Thank you for your valuable comment. We have studied the intermediates during degradation and the products in 60 min using UPLC-MS. The intermediates of N,N-diethyl-N-ethylrhodamine (DER, m/z=415), N-ethyl-N-ethylrhodamine (EER, m/z=387), N,N-diethylrhodamine (DR, m/z=387), and Me-rhodamine (MR, m/z=290) were detected during the degradation process. The intermediates further decomposed to 9-phenyl-3H-xanthene (m/z=258.1), 9-phenyl-1,3a-dihydrocyclopenta[b]chromene (m/z=246) and 2,3-dimethyl-4-phenylchromane (m/z=238). Then, the intermediates generated to organic matter with low molecular weight and less toxicity such as phthalic acid (m/z=166.0), benzoic acid (m/z=122.0), hepta-2,4,6-trien-1-ol (m/z=110.1), and oxalic acid (m/z=90.0). At last, the small molecular acid and alcohol mineralized to CO2 and H2O. However, in our experiment, the intermediates of can still be detected in the product at 60 min, meaning the total mineralization is not finished yet. According to the study of Shi et al. (X. Shi et al., Journal of Colloid and Interface Science, 2022, 610:751-765), the developmental toxicity and bioaccumulation of most intermediates are less than that of RhB. So their overall toxicity in the system gradually decreases during the whole degradation process.

Comments 2: Did you characterized the spent catalyst by EPR spectroscopy ? The EPR is very sensitive and useful for Mo5+, Ni2+, Co2+ species etc.....

Response 2:  Thank you for your valuable advice. EPR is a sensitive tool to analyze Mo5, Ni2+, and Co2+ species. But for Co species, only Co2+ could be detected by EPR owing to its unpaired electron (W. wen et al., ACS Catalysis, 2022, 12, 7037−7045). So we validated the synergistic redox cycles of Co2+/Co3+, Ni2+/ Ni3+, and MO4+/Mo5+ by XPS analysis.

Comments 3: Please give more informations about the EPR methodology for EPR experiments

Response 3:   A solution containing 0.1 g/L of the catalyst was stirred continuously, followed by the addition of PMS (0.5 g). The time was recorded as t0. At specific intervals, 0.5 mL of the solution was withdrawn and filtered through a polytetrafluoroethylene syringe filter disk into a vial. Then, 5,5-dimethyl-1-pyrroline N-oxide (DMPO, 100 mM) was added to capture SO4•- and OH, while, 2,2,6,6-tetramethyl-4-piperidine (TEMP,100 mM) was used to capture 1O2. The mixed liquid was immediately transferred into a quartz tube and inserted into the EPR sample chamber.

Comments 4:  Have you carried out a statistical analysis to determine the average length of the nano-sheets and their average stacking? Do you think that there is a relationship between this parameter and the degradation of Rhodamine B ?

Response 4:  Thanks for your suggestion. We are sorry that we didn’t carry out a statistical analysis to determine the average length of the nano-sheets and the stacking. There should be a relationship between the average sizes and the degradation performance. The size of nanoparticles plays a crucial role in their performance, because it is related with the active reaction sites and the mass transferring between contaminant and catalyst. Your suggestion proposes us a new insight into the future study on the relationship of particle size and the degradation performance.

Round 2

Reviewer 2 Report

Comments and Suggestions for Authors

The manuscript has been corrected according to the comments.